

# Low-regret Climate Change Adaptation in Coastal Megacities – Evaluating Large-Scale Flood Protection and Small-Scale Rainwater Detention Measures for Ho Chi Minh City, Vietnam

Leon Scheiber[1], C. Gabriel David[2], Mazen Hoballah Jalloul[1], Jan Visscher[1], Hong Quan Nguyen[3,4], Roxana Leitold[5], Javier Revilla Diez[5] and Torsten Schlurmann[1]

[1] Ludwig-Franzius-Institute for Hydraulics, Estuarine and Coastal Engineering, Leibniz University Hannover, 30167 Hannover, Germany
[2] Leichtweiß-Institute for Hydraulic Engineering and Water Resources, Division of Hydromechanics, Coastal and Ocean Engineering, Technische Universität Braunschweig, 38106 Braunschweig, Germany
[3] Institute for Circular Economy Development, Vietnam National University – Ho Chi Minh City, 700000 Ho Chi Minh City, Vietnam
[4] Institute for Environment and Resources, Vietnam National University – Ho Chi Minh City, 700000 Ho Chi Minh City, Vietnam
[5] Institute of Geography and Global South Studies Center, University of Cologne, 50923 Cologne, Germany

*Correspondence to*: Leon Scheiber (scheiber@lufi.uni-hannover.de)

**Abstract.** The risk of urban flooding is a major challenge for many megacities in low elevation coastal zones (LECZ), especially in Southeast Asia. Here, the effects of environmental stressors overlap with rapid urbanization, which significantly aggravates the hazard potential in these regions. Ho Chi Minh City (HCMC) in Southern Vietnam is a prime example of this set of problems and therefore a meaningful study case to apply the concept of low-regret disaster risk adaptation as defined by the Intergovernmental Panel on Climate Change (IPCC). In order to explore and evaluate potential options of hazard mitigation, a hydro-numerical model was employed to scrutinize the effectiveness of two adaptation strategies: (1) a large-scale flood protection scheme as currently constructed in HCMC and (2) the widespread installation of small-scale rainwater detention as envisioned in the framework of the Chinese Sponge City Program (SPM). A third adaptation scenario (3) assesses the combined implementation of both approaches (1) and (2).

From a hydrological point of view, the reduction of various flood intensity proxies suggests that the effectiveness of large-scale flood protection outweighs that of small-scale rainwater storage by far. For example, an assessment of the Normalized Flood Severity Index (NFSI) suggests a potential flood reduction that is 3.5 times higher for a classic infrastructure solution than for the Sponge City approach. In contrast, the number of manufacturing firms that are protected from risk after the implementation of disaster risk adaptation significantly excels for the latter response option: while the ring dike mitigates flooding at about 20 % of all considered locations, the assumed rainwater detention would protect up to 93 %. And also, from a governance perspective, decentralized rainwater storage conforms better to the low-regret paradigm: while the large-scale ring dike depends on a binary commitment (to build or not to build), decentralized small- and micro-scale solutions can be implemented gradually (through targeted subsidies) and add technical redundancy to the overall system. In the end, both





strategies are highly complementary in their spatial and temporal reduction of flood intensity, so that local decision-makers
may specifically seek multi-faceted strategies, avoiding singular approaches and designing adaptation pathways in order to
successfully prepare for a deeply uncertain future.

*Keywords*: climate change adaptation, disaster risk reduction, low-regret measures, Ho Chi Minh City, Southeast Asian
megacities, low elevation coastal zones, flood protection, ring dike, rainwater detention, Sponge City

**1 Introduction**

Between 1980 and 2009, the occurrence of floods caused more than half a million deaths and affected another 2.8 billion
people worldwide (Doocy et al., 2013). These figures will further increase with global sea levels rising as a consequence of
Climate Change (IPCC - SROCC, 2019) and more than half of all urban agglomerations (>100.000 habitants) located closer
than 100 km from the coastline (Barragán and Andrés, 2015). Especially in Southeast Asia, where many megacities lie in low-
elevation coastal zones (LECZ), the risk of storm surges will significantly increase (McGranahan et al., 2007; Neumann et al.,
2015; Kulp and Strauss, 2019). And not only will these regions be affected by more frequent extreme storm events, but rainfall
volumes will also increase as a consequence of the Urban Heat Island (UHI) effect fuelled by sustained urban growth (IPCC,
2021). Ranked fifth place in a global assessment of future exposure to climate extremes, Ho Chi Minh City (HCMC; Figure
1: a & b) is a prime example of the complex physical and social interactions that exacerbate the flood risk in many Southeast
Asian metropolises (Hanson et al., 2011; Hallegatte et al., 2013; Abidin et al., 2015). Being the largest city in Vietnam and its
major economical hub, HCMC is subject to uncontrolled urban sprawl, which deepens the exposure to flooding for a growing,
and oftentimes highly vulnerable population (Huong and Pathirana, 2013; Duy et al., 2018). Flood exposure, in this context,
results from the increasing number of settlements in low-lying areas (cf. Figure 1 c), which continuously expand due to the
man-made problem of land subsidence resulting from soil compaction and groundwater exploitation (Kaneko and Toyota,
2011; Erkens et al., 2015; Duffy et al., 2020).
In conjunction with globally rising sea levels and amplified tidal ranges in the adjacent Sai Gon River, this setting leads to
backwater effects in the antiquated and, in many regards, deficient drainage system of HCMC causing widespread inundations
(Phi; Downes and Storch, 2014; Tran Ngoc et al., 2016; MONRE, 2016). Apart from the immanent social implications, urban
flood events cause major and frequent impediments to the local economy, which is estimated to cause losses of USD 48.3
billion of the gross domestic product (GDP) in the period from 2006 to 2050 (ADB, 2010). Especially manufacturing firms
(owning, among others, lots of immovable machinery and facilities) are expected to bear the brunt of damage and loss due to
environmental hazards (Neise et al., 2018). Already in 2015, studies suggested that more than 15 % of the manufacturing firms
in HCMC were located in current or future inundation areas, many of which being small and medium-sized enterprises (SMEs)
that engage about 37 % of all national employees (Leitold and Revilla Diez, 2019). Even if the city has prospered for several
decades and flood control is no longer a question of preventing casualties due to extreme events, it is those recurring floods in
particular that need to be accounted for by designing and implementing suitable adaptation strategies.



**Figure 1: STUDY AREA** – **(a) & (b) Located in proximity to the Mekong Delta, Ho Chi Minh City (HCMC) is the largest city in Vietnam and a major economical hub of Southeast Asia. Administrative boundaries and waterbodies derived from © OpenStreetMap contributors 2022, distributed under the Open Data Commons Open Database License (ODbL) v1.0. (c) Informal settlements are an epitome of the uncontrolled urban sprawl that exacerbates flood risk in this region due to an interplay of increased natural hazards, exposure and vulnerability. Picture taken by the corresponding author in September 2019.**




According to the Intergovernmental Panel on Climate Change (IPCC), there are, in general, five categories of possible adaptation to future risks: retreat, protection, accommodation, advance and ecosystem-based adaptation (IPCC - SROCC, 2019). These strategies can also be differentiated into classic "grey" infrastructure and nature-based "green" solutions (Dong et al., 2017; Morris et al., 2018). One recent example of the latter approach is the Chinese Sponge City program (SCP), a framework which refines established concepts like Water-Sensitive Urban Drainage (WSUD) and Low Impact Development (LID) (Jia et al., 2017; Qi et al., 2020; Sun et al., 2020; Li and Zhang, 2022). To address the predicament of increasing natural hazards in more and more urbanized areas, Sponge Cities make use of the natural hydrological cycle to effectively reduce urban flooding, harvest rainwater and improve water quality as well as to restore ecological values (Köster, 2021; Jia et al., 2022). In practice, this can be accomplished by medium- and small-scale elements that allow water storage (detention basins and rain tanks), infiltration (permeable pavements and infiltration wells) or both (public parks and rooftop/rain gardens) (Nguyen et al., 2019; Kumar et al., 2021). Since its introduction to the public in 2013, the Sponge City program has shown its feasibility for multiple real-world examples across China (Hawken et al., 2021; Yin et al., 2022), yet, especially its adaptability and possible small-scale implementation make it highly attractive to other rapidly growing megacities in Southeast Asia. Nevertheless, the aim of both grey and green adaptation strategies should be to offer a solution that is of a low regret. In the context of the IPCC, the term "low-regret" does not necessarily refer to financial aspects, but rather aims at solutions that allow coping with current challenges and hazards without impairing future options to deal with climate change effects (IPCC, 2012). In turn, actions that alleviate current adaptation needs, but have a negative effect on future adaptation pathways are "maladaptive" (IPCC, 2014). From a socio-political perspective, adverse or deliberately false decisions leading to (recurrent) maladaptive actions are called maldevelopment (David et al., 2021). Maldevelopment is followed by continuously degrading conditions with the risk of reaching and exceeding anthropogenic tipping points – critical thresholds, from which changes to a system cannot be reversed (Duvat and Magnan, 2019). Given these significant consequences of actions and the wide range of climate change projections, suitable means of adaptation are preceded by robust decisions. With regard to low-regret adaptation, robust decision-making does not aim at providing solutions for all future scenarios, but rather lays the foundation and does not impair to account for a variety of possible impacts (Marchau et al., 2019). In this context, climate adaptation builds on a more flexible strategy including scenario-based pathways and an "agree-on-decisions" approach rather than the more traditional "predict-then-act" approach (Lempert, 2019). With regard to flooding in Southeast Asian metropolises, low-regret flood adaptation should be based on multiple options, which consider the trajectories of natural hazards, like sea-level rise and the concentration of extreme rainfall, on the one hand, and anthropogenic factors, such as uncontrolled urban growth and man-made land subsidence, on the other.

For the particular case of HCMC, few authors investigated the potential of available, grey or green, flood adaptation options within the last years. Most prominently, the Vietnam Climate Adaptation PartnerShip (VCAPS) consortium presented an extensive flood protection scheme that comprises various, mostly classic adaptation measures (VCAPS, 2013). Supported by multiple flood gates and five major pumping stations, a large-scale ring dike prevents water from the Sai Gon River from entering the inner-city canal system and, thus, literally decouples the city centre from the natural hydrological regime of the





Sai Gon River. On this basis, annual damages for five different extreme events under the aforementioned adaptation strategy could be estimated by implementing the individual infrastructural elements into a hydro-numerical model of one particularly flood-prone area (Lasage et al., 2014). A comprehensive evaluation of the suggested adaptation measures and their
combination was subsequently published modelling present and future storm surges for the complete city area and providing a detailed cost-benefit analysis (Scussolini et al., 2017). The study concludes with a map of proposed adaptation pathways for Ho Chi Minh City. Construction of the ring dike was, in the meantime, ordered by the Vietnamese Ministry of Agriculture and Rural Development (MARD) in 2016. Besides infrastructural flood responses, also the importance of governance decisions such as reasonable land-use planning was emphasized, to minimize the exposure and allow for the vulnerability of the ever-
growing urban population (Downes and Storch, 2014). Another recent study focused on surveying and discussing examples of ecosystem-based flood adaptation in terms of Sustainable Urban Drainage Systems (SUDS) (Ho et al., 2015).

In summary, nearly all suggested flood adaptations for HCMC originate from recent work of the VCAPS consortium and their effectiveness has been thoroughly tested for the case of extreme storm surges with return periods of up to 1000 yr. However, hardly any research has been conducted to evaluate the performance of the proposed grey solutions, such as the ring dike, or
any ecosystem-based approaches with respect to typical "every-year" events, which is in stark contrast to their hundredfold estimated GDP loss expected by the Asian Development Bank (ADB, 2010). Moreover, heavy rainfall has been completely disregarded in the majority of studies, although precipitation-based damages should demonstrably be an essential part of all urban flood risk assessments (Rözer et al., 2019). Finally, only a few working groups have scrutinized the combination of grey and green flood adaptation measures (Hamel and Tan, 2021). To address this gap in current research, the presented study
scrutinizes the hydraulic effectiveness of grey and green flood adaptation measures under the influence of a representative flood event, i.e. tidal water levels between -1.6 m and +1.2 m above sea level in combination with the official design storm (T = 1 a; D = 180 min). By employing a simple, yet robust surface runoff model, the focus is set on the following objectives:

- To explore the feasibility and juxtapose the hydraulic effects of two exemplary grey and green flood adaptation measures given the case of HCMC,
- To quantify the potential reduction of flood intensity, expressed through conventional and integrated proxies, that is achievable by each adaptation strategy and their combined implementation,
- To evaluate their low-regret character (including tentative co-benefits) and discuss the implications for decision-makers.

Classic, grey adaptation measures in this study are following the concept of the large-scale ring dike as envisioned by VCAPS.
Decentralized, green flood adaptation, on the other hand, is represented by small-scale rainwater detention as realized in the context of the Chinese Sponge City program. Not only do these two approaches differ in terms of their size and hydraulic working principles, but the large-scale ring dike is also characterized by its binary realization, while the Sponge City concept is inherently modular. It is the aim of this study to juxtapose these two strategies both from a hydrological and a governance point of view elucidating the practical validity of the IPCC's low-regret paradigm.



## 2 Material and Methods

To assess the hydraulic effectiveness of various adaptation strategies, a simplified two-dimensional flow model, designed in HEC-RAS Version 6.0 by the U.S. Army Corps of Engineers (USACE, 2021), was used to simulate rainwater-induced inundations in HCMC. The key principle behind the employed modelling scheme is that precipitation volumes typically accumulate across the urban topography and finally discharge into the local canal system (physically-based rainfall-runoff modelling). The software platform was first introduced in 1995 and has been validated in numerous case studies in comparable settings ever since (Patel et al., 2017; Muthusamy et al., 2019; Rangari et al., 2019; Yalcin, 2020). As a consequence of local data availabilities, the complete setup was based on open-access input data in the form of freely available topography and hydro-meteorological conditions. The computational grid comprises about 17.5 Mio rectangular cells of ca. 85 m x 85 m resolution with combined elevation data from the Shuttle Radar Topography Mission (SRTM, 2007) and the CoastalDEM (Kulp and Strauss, 2018). On this basis, the model domain is defined by the local terrain as it contains all catchments that potentially contribute to surface runoff within the city boundaries (cf. Figure 2 a). Upstream influxes are given by the long-term mean discharges of Sai Gon (MQ = 54 m³s-1) and Dong Nai (MQ = 890 m³s-1) rivers, respectively (Tran Ngoc et al., 2016). The downstream boundary condition in the south of the city was obtained by extrapolating coastal water levels from a global database (Caldwell et al., 2015) in accordance with hydrological reports for this catchment (Gugliotta et al., 2019). The model boundaries are complemented by uniformly distributed rainfall with a hyetograph that follows an official three-hour design storm and can be readily adapted for arbitrary return periods (cf. Figure 2 b). As there is hardly any public information about the quality and condition of the existing drainage system in HCMC – except that its capacity is regularly overloaded during extreme events (Scussolini et al., 2017) – all rainfall is assumed to become gradient-controlled surface runoff. The resulting flow paths resemble the course of the physical drainage system, including site-specific backwater effects, so that reported inundation hotspots could be reproduced reliably. After calibration, the runoff model was validated against locally reported inundations from 25 different locations across HCMC (documenting a storm event in June 2010). The comparison of simulated and observed flood depths yielded a root mean square error of RMSE = 0.03 m and a correlation coefficient of $R^2$ = 0.75, respectively, making the modelling approach trustworthy enough for all intended tasks, i.e. the direct comparison of flood intensities under different adaptation scenarios.

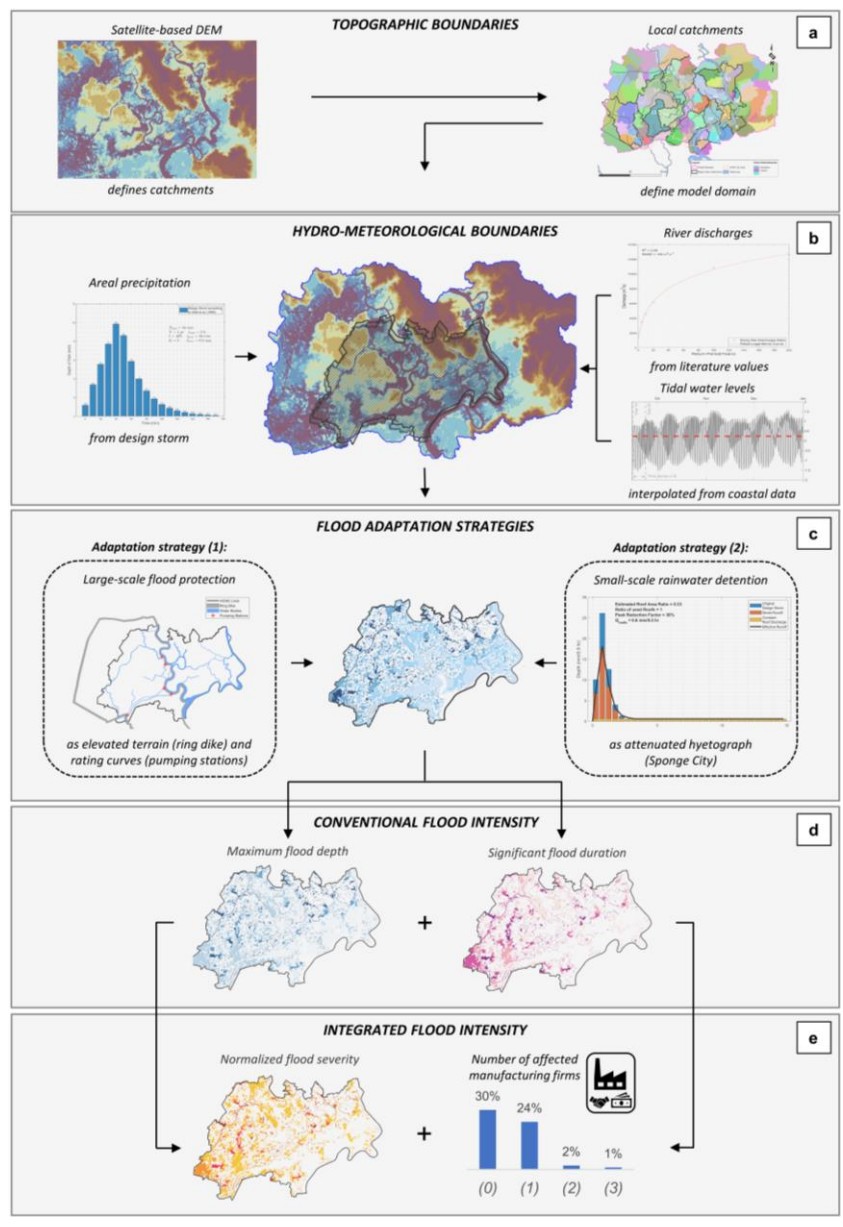

**Figure 2: MODEL SETUP** - A bias-corrected digital elevation model was derived from the combination of SRTM and CoastalDEM; the local catchments that result from the local topography define the modelling domain (a: Topographic boundaries). While tidal water levels were extrapolated from coastal data, average river discharges were taken from the literature and precipitation was forced on the model boundary in accordance with the valid design storm focussing on a return period of one year (b: Hydro-meteorological boundaries; all elevation data displayed using scientific colour maps (Crameri, 2022)). Two adaptation strategies were investigated by either implementing changed elevations due to a large-scale ring dike in combination with five parametrized pumping stations (1) or reproducing the hydraulic effects of decentralized water detention according to a Sponge City approach in the form of an attenuated hyetograph (2), respectively (c: Flood adaptation strategies). The flood simulations resulted in spatio-temporal data on water surface elevations revealing local maximum flood depths and durations of significant flooding (d: Conventional flood intensity). Finally, the local flood severity was estimated via the Normalized Flood Severity Index (NFSI) and in terms of manufacturing firms that were still affected by significant flooding after implementation of the individual adaptation strategies (e: Integrated flood intensity; all numerical results displayed in monochromatic colours for illustration purposes).





The described surface runoff model was employed to assess four different adaptation scenarios:

180         (0) a base case scenario without any technical adaptation to mitigate urban flooding,

        (1) a large-scale protection project including a ring dike, multiple sluice gates and pumping stations

        protecting the central districts of HCMC from storm surges (and, in parts, pluvial flooding)

        (2) a small-scale rainwater detention scheme as implemented in the Sponge City program,

        mitigating the peak of rain-induced surface runoff and

185         (3) a combined application of strategies (1) and (2).

Apart from a reference simulation without any adaption options (0), the first strategy (1) originates from the large-scale development project envisioned by VCAPS that was initiated by the Vietnamese Ministry of Agriculture and Rural Development (MARD) in 2016. In the framework of the numerical model, the central ring dike was implemented by locally fitting digital elevations to a designated height of 2.3 m above sea level (cf. Figure 2 c left & Figure 3 c), thus, directly affecting

the simulated surface runoff in this area. The pumping stations, on the other hand, were realized by a customary HEC-RAS feature (since Version 6.0), which allows cell-specific discharges according to a user-defined rating curve. The five pumping stations were placed vis-à-vis of the flood gates, i.e. at all crossings between the ring dike and a major intercity canal (cf. Figure 2 c left & Figure 3 c). In contrast, the six large-scale sluice gates (and multiple smaller ones) that are also part of the underlying "MARD plan variant" (Phi et al., 2015) were not implemented in this study as these facilities would be locked and

therefore without hydraulic effect in the considered flood events. While the structural solutions of the first adaptation strategy were integrated directly into the hydraulic model, the decentral approach of the second strategy (2) had to be implemented in a parameterized manner. All rainwater detention measures – be it public basins, green roofs or private rain barrels – generally procure a similar alteration of surface runoff: while the effective volume stays the same, the local maxima in the hydrograph are attenuated and converted into (throttled) discharge of manageable magnitude and time. According to this volumetric

analysis, the comprehensive application of small-scale rainwater detention as a realization of the Sponge City concept was represented by directly changing the hyetograph of the regular design storm, thereby translating the hydraulic effect of decentral water storage into attenuated and prolonged rainfall (cf. Figure 2 c right). In this connection, an approximate assessment based on Sentinel-2 images revealed that about one-third of the included area is covered by roofs. Optimistically estimating the potential of household-level rain detention, it was assumed that a maximum of one half of this area could be

utilized for this type of water storage leaving about 15 % of the total area to be considered in the parametrization scheme. The remaining scenario (3) finally combines the large-scale ring dike project with small-scale rainwater detention in order to assess the complementarity of these seemingly adverse strategies.

All simulations yield spatio-temporal output data describing the water surface elevation at every cell of the modelling domain over time. To understand the resulting risks, flood intensity was further assessed through four proxies in two steps: the flood

height (Figure 2 d, left), which quite intuitively represents time-independent maximum inundation depths at every point and the flood duration (Figure 2 d, right) defined by the time span a certain point experiences a significant inundation, i.e. a depth greater than 10 cm. These two parameters are the most common proxies for flood intensity and can directly be derived from





the model outputs. Secondly, a more integrated view on the potential reduction of flood risk is provided by the Normalized Flood Severity Index (NFSI) (Figure 2 e, left) and the number of affected manufacturing firms (Figure 2 e, right). The NFSI

combines flood depth and duration into one dimensionless parameter and is calculated as the normalized product of these two conventional flood intensity proxies. Accordingly, this easy-to-apply index emphasizes those areas, which are exposed to significant flooding over a significant time and, thus, are expected to experience the most severe damage across a given study area.  A detailed explanation of the rationale behind the NFSI and its mathematical definition can be found in an independent publication (Hoballah Jalloul et al., in review). The location of manufacturing firms in HCMC is based on information from

the Vietnam Enterprise Census 2017 collected by the General Statistics Office of Vietnam. This data were previously utilized in a study, which investigated the constraints that affect firm decisions in undertaking flood adaptation measures based on SRTM elevation data (Leitold et al., 2021). In contrast to traditional cost-benefit analyses that require sophisticated damage/loss models, the avail of assessing only the number of affected firms is that all companies are valued equally, thereby avoiding an underestimation of (non-monetary) SME values (Kind et al., 2020; Hino and Nance, 2021). On this basis, the

hydraulic effectiveness of each adaptation strategy was expressed through difference maps highlighting the local reduction of each of the described proxies in comparison with the base case, allowing a qualitative estimation of the flood reduction that each of the hypothetical strategies offers, and, what is more, facilitating a direct juxtaposition of their hydraulic effectiveness.

## 3 Results

Allowing for the highly dynamic behaviour of surface runoff volumes across the HCMC model, all spatio-temporal output

data were converted into heat maps for the first three flood intensity proxies. The potential flood reduction of a given adaptation strategy was then illustrated by subtracting the respective base case heat maps. The benefits of this methodology become clearer when studying the distribution of flood severity reductions as depicted in Figure 3. In particular, the upper two panels (a) and (b), corresponding to strategies (1) and (2), visualize the reduction in flood severity due to the implementation of the VCAPS ring dike (incl. sluice gates and pumping stations) and the comprehensive application of water detention, respectively.

When comparing the two strategies, areas with reduced flood severity (highlighted in green colour) significantly exceed after implementation of the first adaptation strategy (Figure 3 a), although they form certain clusters along the low-lying areas close to the major canals. This seems obvious given that the dike eradicates any tidal influence and the heavy-duty pumps are capable of emptying these crucial elements of urban drainage within hours. Then again, there are several areas (highlighted in red colour) that experience increased flood severity specifically across the western, riverine parts of the municipal city of Thu Duc

and in several rural areas across the east of HCMC. Flood reductions for the second adaptation strategy (Figure 3 b), on the other hand, are less pronounced, but by far more uniformly distributed likewise addressing low- and high-lying areas. In addition, this strategy does hardly induce any worsening with regard to flood severity except for the exact spots that are alleviated by the ring dike strategy.





Figure 3: SIMULATION RESULTS - Spatial reduction of flood intensity in terms of decreases in normalized flood severity (NFSI).
(a) While the large-scale ring dike strategy improves the flooding situation in the low-lying areas along the major canals, but worsens
the situation for the western parts of the municipal city of Thu Duc, (b) the decentralized water detention scheme shows more
uniformly distributed improvements. (c) The combination of both approaches yields even better results as increases and decreases
in flood severity overlap for the former two strategies in a highly complementary way. All NFSI reductions displayed using scientific
colour maps (Crameri, 2022).





Changes in flood severity that can be expected for a combination of both strategies are depicted in the lower panel (Figure 3 c). At first sight, most flood severity reductions visible in (a) can also be found for the combined approach in (c). When looking at the increases in flood severity, however, several of the inundations, which are induced by the implementation of a ring dike, diminish in case of additional rainwater detention. This applies particularly to the areas east of the Sai Gon River

suggesting that certain detrimental effects of excessive pumping could be mitigated by a combined adaptation strategy. Nevertheless, the spatial changes are hard to differentiate from visual inspection alone.

Adding to these findings, the quantitative results underline that both grey and green adaptation strategies are, in general, capable of alleviating the average intensity of local flooding in HCMC. This can be concluded, for instance, from average values of maximum flood depths and flood durations, which are significantly reduced in relation to the reference case without

adaptations (Table 1). In direct comparison, however, it becomes obvious that the implementation of (1) a ring dike (incl. flood gates and pumping stations) would reduce average flood depths about two times and flood durations even nine times more effectively than (2) widespread water detention. A closer look at the potential improvements from (3) a combined approach suggests reductions of flood depth that are even higher than the sum of the single components of strategies (1) and (2).

**Table 1: Potential reduction of flood intensity for three adaptation strategies: (1) Large-scale ring dike scheme including pumping stations, (2) comprehensive application of decentralized rainwater detention and (3) the combination of both approaches. Improvements are expressed through decreasing mean values of the conventional flood intensity proxies, maximum flood depths in cm (left) and significant flood duration with depths higher than 10 cm in hours (right), in comparison with respective values for (0) the base case without adaptation measures.**

| Strategy | Conventional Flood Intensity Proxies | | | |
| --- | --- | --- | --- | --- |
| | Flood depth ($d_{max}$) | | Flood duration ($T_{d>10\,cm}$) | |
| | (cm) | (%) | (h) | (%) |
| (1) Large-scale protection | -2.7 | -19.5 | -9.2 | -44.2 |
| (2) Small-scale detention | -1.5 | -10.6 | -1.1 | -5.5 |
| (3) Combination | -4.8 | -34.5 | -9.2 | -44.4 |

Although this does not hold true for flood duration and affected firms, the numerical simulations still reveal that the hydraulic benefits of a combined approach always outmatch individual solutions, which is further corroborated by the other, more integrated flood intensity proxies (Table 2). For example, the NFSI for adaptation strategies (1) and (2) is reduced by -41.7 % for strategy (1) and -11.9 % for strategy (2), respectively, in comparison to the base case without adaptation. This translates into a 3.5-times higher potential of flood reduction for the classic infrastructure solution (1) than for a Sponge City approach

in the outlined framework (2). The combination of both would finally yield a 46.7 % improvement in NFSI values. A different picture emerges, when assessing how the different adaptation strategies influence the number of manufacturing firms, which experience a significant flood depth of more than 10 cm. While the classic strategy (1) only protects 19.9 % of the listed





companies, decentralized detention measures (2) could reduce the current number by 93.3 %. This again translates into a 4.7-times more effective investment in the case of strategy (2). For a combined approach of classic protection and rainwater storage,

this even corresponds to a 4.9-times higher effectiveness compared to the ring dike alone. However, although these numerical results indicate that small-scale measures can support future flood adaptation concepts in an effective and straightforward manner, they do not take into consideration the low-regret qualities of the assessed options and the implications that arise for local decision-makers.

**Table 2: Potential reduction of flood intensity for three adaptation strategies: (1) Large-scale ring dike scheme including pumping stations, (2) comprehensive application of decentralized rainwater detention and (3) the combination of both approaches. Improvements are expressed through decreasing values of the integrated flood intensity proxies, the average normalized flood severity index (NFSI) (left) and the total number of manufacturing firms that are affected by a flood depth greater than 10 cm (right), in comparison with respective values for (0) the base case without adaptation measures.**

| Strategy | Integrated Flood Intensity Proxies | | | |
| --- | --- | --- | --- | --- |
| | Flood Severity (NFSI) | | Number of affected manufacturing firms | |
| | (units) | (%) | (-) | (%) |
| (1)  Large-scale protection | -7.0 | -41.7 | -146 | -19.9 |
| (2)  Small-scale detention | -2.0 | -11.9 | -683 | -93.3 |
| (3)  Combination | -7.8 | -46.7 | -708 | -96.7 |

## 4 Discussions and Conclusions

Several studies have recently assessed the available flood adaptation options of HCMC, with a special focus on resistance against extreme storm surges and such hazards under future sea-level rise scenarios (VCAPS, 2013; Lasage et al., 2014; Scussolini et al., 2017). However, no attention was paid to the more frequent, especially pluvial events, although these incidents demonstrably cause financial losses that are up to ten times higher than singular extremes (ADB, 2010) and affect small and medium manufacturing firms, which are the backbone of the national economy, to a disproportionately high degree (Leitold

and Revilla Diez, 2019). To address this knowledge gap, a numerical model was employed to evaluate possible responses to urban inundations resulting from heavy precipitation of an annual return period. Two exemplary adaptation measures, (1) a classic (grey) protection scheme in the form of a large-scale ring dike and (2) a (green) Sponge City approach in terms of decentralized rainwater detention, were implemented in a process-based, hydrological model. However, the fact that the model was exclusively built on freely accessible boundary conditions introduces some uncertainties to the setup: first of all, the

elevation data from SRTM and CoastalDEM typically come with a site-specific vertical bias as intensely discussed during the last years (Schumann and Bates, 2018; Kulp and Strauss, 2019; Vernimmen et al., 2020; Mukul et al., 2015). A comparison





with three random LiDAR samples suggests that, also for the specific case of HCMC, errors in the order of 1 m are to be expected (A = 4.26 km²; ME = -1.07 m; STD = 1.09 m; RMSE = 1.52 m) (Hoballah Jalloul et al., in review). Yet, although the results of the here employed model are highly dependent on the topographic conditions across the city as well, the influence

of a vertical bias diminishes, when simulation runs with the same systematic error are directly compared. The same applies to the (conservative) assumption that the local drainage system is malfunctioning from the beginning of each model run. Furthermore, the impact of local land subsidence was not considered in the assessed simulations of present-day adaptation responses, but the extrapolation of such trends should be a requisite for simulations that address options under future climate change projections given their significant share in relative sea-level rise (Nicholls et al., 2021). Although the employed surface

runoff model is based on these assumptions, its implications still go beyond earlier, DEM-based analyses (Dang and Kumar, 2017; Leitold and Revilla Diez, 2019) and its limitations are comparable to other process-based models in this region (Scussolini et al., 2017). Even though simulated urban inundations should not be mistaken with quantitative forecasts, the reduction of flood intensity and especially the spatio-temporal differences between the assessed strategies can help both researchers and decision-makers to critically compare the hydraulic effects of available adaptation options. In this context, the

combination of conventional and integrated flood intensity proxies allows for a holistic view of their potential in physical flood risk reduction and avoids a singular focus on economic assets, which would underestimate the social relevance of small and medium enterprises (Kind et al., 2020; Hino and Nance, 2021).

The obtained simulation results suggest that both adaptation measures are generally capable of alleviating the flooding situation in HCMC, albeit in very different dimensions. The classic protection scheme has a much higher reduction rate regarding flood

intensity than the projected rainwater storage. But then, positive effects of the ring dike are also limited to the low-lying areas in the vicinity of the intercity canals, which are easily drained in contrast to the more remote and high-lying districts. What is more, the grey adaptation measure relies on sufficient capacities and smooth operation of few central pumping stations, which makes this concept highly vulnerable in case of technical or operational failure. The importance of technical robustness and reliability becomes particularly clear when considering the role of the existing drainage system. Outdated and in many regards

deficient, this system is one of the main drivers of growing flood risk in HCMC and an epitome of critical dependence on grey infrastructure and fragile technical solutions. A Sponge City approach, in contrast, is characterized by its modularity comprising a large number of decentralized elements. This is a major advantage as it disperses the risk of failure through redundancy and therefore increases the resilience of the overall system. However, in urban settings, the implementation of medium-scale solutions for rainwater storage, like dedicated detention basins, may entail conflicts with other (mostly

commercial) interests in land use management due to the required consumption of limited public space. Multiple administrative authorities, private land owners and developers typically need to be involved to make decisions. Although arising conflicts can be addressed through active participation by representatives of the local citizens and businesses as well as experts from the administration in a joint co-design process (Chen et al., 2021), their complexity can prolong and in some cases even prevent such implementation plans. Hence, the most promising approach to integrate the Sponge City concept into a highly urbanised

area, like HCMC, would be the promotion of small- and even micro-scale solutions on a neighbourhood and household or firm





level. These are inherently easier to realize than medium-scale solutions as they can make use of existing residential spaces or vacant lots (not being subject to commercial interests) and can be governed through incentives in a much simpler manner. Also, concerning future trends, both strategies differ in their degree of sustainability: because the operation of grey flood protection measures, and specifically the ring dike, is defined by absolute design heights, these structures are constructed for

one specific water level projection and constant land subsidence rates, whereas these two components of relative sea-level rise hardly impair the potential of rainwater detention. Although classic flood protection shows higher hydraulic effectiveness, this study of HCMC shows that small-scale solutions, as projected in the framework of a Sponge City approach, excel in terms of technical resilience and that they are readily available options for addressing residual flood risk. Especially the combination of both strategies offers a much-needed technical redundancy, besides the obvious gains in quantitative flood reduction.

With regard to the low-regret paradigm, flexible approaches and strategic planning for the future are central elements of a successful disaster risk response strategy (IPCC, 2012; Marchau et al., 2019). Typically, an adaptation plan contains multiple adaptation pathways, visualizing the impact and dependence of single adaptation measures on future capacities to deal with climate change effects (Haasnoot et al., 2013; Kwakkel et al., 2015; Haasnoot et al., 2019). Thinking adaptation with a low-regret mindset facilitates considering small-scale measures along the way, with the chance to avoid more drastic interference

within the natural and social setting (David et al., 2021). With this in mind, the ring dike in HCMC is a direct protection measure for the low-lying parts of the inner city. Complemented by heavy-duty pumps, it shields the central districts from harmful storm surges and eradicates the tidal influence on the intercity canals. However, even if parts of the precipitation volumes remain in higher elevated areas, unable to drain through the deficient drainage system, this surplus of discharge still raises the water levels of the gaining Sai Gon River. Like the general dependence on a single technical structure, this fact

shows a maladaptive tendency as it can be detrimental to the outer reaches of HCMC, which are increasingly inhabited by the poorer, more vulnerable portions of the population (Duy et al., 2018), who now experience even higher floods. To alleviate this tendency and prevent maldevelopment, the currently constructed solution can be complemented with incentives for societal and communal flood protection: Decentralized rainwater detention verifiably attenuates the runoff peak and allots discharges to a longer period (Jiang et al., 2022). Considering an adaptation pathway plan for HCMC from a governance perspective, the

ring dike is a unique project based on a binary decision (to build or not to build). It is designed to protect a given domain from a clearly specified set of natural hazards. Rainwater detention, on the other hand, is a modular category of individual measures following the Sponge City concept, which does not need further specification and can be tailored to the specific demands of the local situation. This makes this approach highly flexible as its spatio-temporal implementation can be controlled by targeted subsidies, e.g. for purchasing and installing private rain barrels etc. Such micro-scale solutions enable both individuals and

SMEs, who currently depend on top-down decisions, to actively participate in (bottom-up) climate change adaptation on a community and household level. Finally, this approach tackles flooding at its source and does not interfere with other protection measures on site. These aspects – the flexibility, the capacity of local stakeholders to respond gradually and in accordance with near-future precipitation projections as well as the ability to mitigate without impairing protection efforts on site – make the Sponge City concept a prime addition to a successful low-regret adaptation strategy.





As mentioned before, the construction of a ring dike, multiple flood gates and pumping stations is currently underway in HCMC. This adaptation pathway is an effective response to the current flooding situation as it protects the central districts of the city from high tides and coastal storm surges. However, frequent rain events cause major socio-economic disruptions and induce significant financial losses to the local economy as well. To address those areas that are not yet protected from pluvial flooding or even adversely affected by the ring dike, pursuing a Sponge City approach would be a second and highly

complementary adaptation pathway. In this context, decentralized rainwater detention in any form – be it green roofs, rain barrels or detention basins – can be considered useful, especially in view of the impending rise of relative sea levels through global warming and local land subsidence. This may be even more beneficial as many of these solutions can be implemented as multi-purpose structures or possess valuable ecosystem services, first and foremost beneficial to mitigate Urban Heat Island effects (He et al., 2019). Although further adding to the low-regret character of this concept, an assessment of the co-benefits

of individual Sponge City measures would go beyond the scope of this study. In the end, it is highly desirable to consider this vital concept, when technically approaching the increasingly severe disaster risk of low-elevation coastal zones in general and of Southeast Asian metropolises in particular, to allow them to prepare for a deeply uncertain future.





**Code availability**

No code was used in this research. Details about the general processing of numerical data are provided in the methods section or can be inquired from the corresponding author.

**Data availability**

The references and freely available data used in this study can be accessed through the respective journals or databases.

**Author contribution**

LS, MHJ and JV designed the hydro-numerical model finally set up and operated by MHJ. JRD acquired the census data, from which RL extracted the geo-location of manufacturing firms. MHJ processed the simulation results, which LS, MHJ and JV interpreted. LS and CGD conceptualized the paper outline. LS and CGD wrote the initial manuscript with input from MHJ, while JV, JRD, HQN and TS reviewed and edited the final text. JV and TS (co-)designed the overarching research project, were responsible for funding resources at LuFI as well as JRD at GSSC. All three provided guidance throughout the entire
study.

**Competing interests**

The authors declare that they have no conflict of interest.

**Acknowledgements**

The list of inundation hotspots, forming the basis for the model validation in this study, was generously provided by Dr.
Nguyen Quy from the HCMC-based engineering company EPT Ltd. Moreover, the authors wish to express their gratitude towards Friedrich Hilgenstock from WTM Engineers GmbH for his expert guidance in technical flood adaptation options. Finally, sincere thanks go to the editor at NHESS for handling the manuscript and two anonymous reviewers for their helpful comments.

**Financial support**

This research has received funding from the DECIDER project sponsored by the German Federal Ministry of Education and Research (BMBF; grant no. 01LZ1703H).



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
