# Peer review of "Low-regret Climate Change Adaptation in Coastal Megacities – Evaluating Large-Scale Flood Protection and Small-Scale Rainwater Detention Measures for Ho Chi Minh City, Vietnam"

_Natural Hazards and Earth System Sciences, 2022_

## Author Comment (AC1)

RESPONSE TO REVIEW #1

We highly appreciate time and effort the reviewers put into studying and reviewing our manuscript, thank you for initiating this exchange! After carefully reading and discussing the remarks, we have thoroughly revised and improved the manuscript accordingly. Please find our responses (blue) and revised text blocks (*blue, italic*) below the quoted reviewer comments (**black, bold**). Please note that, in the meantime, we had the invaluable opportunity to visit the study site in person and subsequently incorporated newly gathered ground truth data in the numerical modelling scheme. Moreover, we included a few minor changes that might not refer directly to specific reviewer comments, but are meant to enhance the readability and hence understanding of our approach and findings according to a native speaker.

**General Comments**

**This paper compares the effectiveness of two adaptation strategies: (1) a large-scale flood protection scheme as currently constructed in HCMC and (2) the widespread installation of small-scale rainwater detention as envisioned in the framework of the Chinese Sponge City Program (SPM). As authors claimed, it is important to explore and evaluate potential options of hazard mitigation as flood risk is becoming a major challenge for many cities in low elevation coastal zones. The topic of this study is valuable. But the quality and innovation of the current manuscript are not satisfactory. In any case, I have a few recommendations that I believe will help the authors to clarify their contribution and improve the readability of the manuscript.**

We are very happy that the reviewer acknowledges our motivation and the intended value of the presented topic. This statement confirms our original idea to present and disseminate our simple, but generic approach to explore and evaluate hazard mitigation option to alleviate urban flood risks in low elevation coastal zones and our specific findings from HCMC to a wider audience. We are in line with the reviewer's perspective that flood exposure studies and tailor-made adaptation measures become gradually more important, especially to flood-prone communities in emerging economies and developing countries. To improve the quality and innovation of the current manuscript, we have addressed and diligently integrated the reviewer's queries and recommendations. The provided comments, indeed, helped us a lot to clarify the objectives and current limitations of our contribution and improved the readability and quality of the manuscript. Details about the specific revisions undertaken can be found below.

**Specific points are:**

1. **Most of the figures in the manuscript are very poor in quality and hard to meet the standard for this journal, such as Figs. 2, not clear enough. Generally, some of the figures are too small.**

   This feedback is very helpful as it seems that illustrations originally saved in 600 dpi resolution were compressed and therefore blurred during PDF conversion. We will make sure that all figures will be provided in original quality in the revised manuscript. Based on this comment, we also revisited mentioned Fig. 2 and decided to simplify some of the depicted model in- and outputs for a more intuitive perception of the illustrated workflow.

2. **How to simulate the small-scale rainwater detention in the HEC-RAS model? What is the limitation or uncertainty?**

   We are happy that the reviewer pointed out this lack of clarity. Following the previous comment, we tried to improve the readability of figures including our illustration of the attenuated hyetograph (Fig. 2 c right), which allows us to simulate small-scale rainwater detention by means of a parametrization. We hope that, accordingly, the explanations given in section 2 Material and Methods (ll. 207-216) should now give a sufficient explanation of the rationale behind this way of implementation. Instead, we revised section 4 Discussions and Conclusions (ll. 309-318) adding further discussion of the limitations/uncertainties inherent to this approach. The paragraph now reads as follows:

   *"Additional limitations arise from the parametrization of rainwater detention in the form of an attenuated hyetograph (cf. section 2 Material and Methods) (…) The estimation of the roofed area from satellite imagery and corresponding detention capacities entails additional uncertainties with respect to the assumed runoff attenuation. In reality, the actual shape of this hyetograph depends on a multitude of factors including technical details about the individual solutions (how much storage volume per unit) as well as the degree of implementation (how many units per area). Nevertheless, the presented approach is sufficiently descriptive for a conceptual juxtaposition of the effects and performance of the two mitigation strategies under consideration and demonstrates the general working principle despite the underlying simplifications: (…)"*

3. **I am not convinced by the model setup given limited information, more information for the validity of flood simulation by HEC-RAS should be described in the paper.**

This comment addresses a concern, which has already been much-discussed within our group before submission. In the current version of the manuscript, explanations regarding the numerical setup were deliberately kept short to be readily accessible for non-experts in the field of hydraulics and water resources. The same group of authors currently aims to publish, in parallel, the more detailed aspects of the flood simulation in a separate manuscript submitted by Hoballah Jalloul et al. (in review) in the same Special Issue of NHESS. This second manuscript not only addresses all specificities about the acquisition and processing of input data and the calibration/validation of the model, but also elaborates on the general validity and performance of open-access data in any numerical flood risk analyses. The methodology is meant to enhance understanding and build capacities to create sound adaptation strategies. In order to keep the focus and objectives of our work as distinct as possible, we decided to separate the two studies and consequently reduced methodological details in the current manuscript to a minimum. Nevertheless, we added an explicit in-text reference here to improve the balance between both publications. This allows readers with a modelling background to follow up on the methodology, while maintaining a risk management perspective within this study. The description of the model setup (referring to Fig. 2 b) has been revised and now reads as follows:

*"For a more detailed explanation regarding the processing of input data as well as the calibration/validation of the employed model, please be referred to the independent publication by Hoballah Jalloul et al. (in review), which discusses the general validity of open-access data in numerical analyses more profoundly."*

4. **What is "Flood Severity Index (NFSI) ", how is it measured?**

Similar to the previous comment, we acknowledge the lack of clarity with regard to the Normalized Flood Severity Index (NFSI) as a consequence of our attempt to balance the readability for non-experts in the field by simplifying technical terms. We also understand that readers should get all information that are relevant to understand the implications arising from the definition of this variable and its significance for risk assessment. Accordingly, we added the following mathematical definition of the NFSI to this manuscript:

*"For a given pair of coordinates, the NFSI is calculated as the product of these two conventional flood intensity proxies divided by the product of the 95 % percentiles of the same proxies as follows:*

$$NFSI(x, y)(\%) = \frac{z_{max}(x, y) * DoT(x, y)}{z_{95\%}(x, y) * DoT_{95\%}(x, y)} * 100 \qquad (1)$$

*where $z_{max}(x, y)$ refers to the maximum simulated flood depth at coordinates x and y and $DoT(x, y)$ refers to the inundation duration over the pre-defined threshold at the same coordinates. As a qualified first estimate, this easy-to-apply index emphasizes those areas, which are exposed to significant flooding over a significant time and, thus, are expected to experience the most severe damage across a given study area."*

For further details, an in-text reference should now direct the interested reader to the companion paper by Hoballah Jalloul et al. (in review), which further elaborates on the methodological details:

*"A detailed explanation of the rationale behind the NFSI and its validation can be found in an independent publication by Hoballah Jalloul et al., (in review)."*

**5. Drainage capacity should be considered for the flood simulation.**

The reviewer addresses an important limitation, here. We generally agree that a direct implementation of this detail would be highly desirable, as it could add further credibility to the numerical setup and robustness to subsequent calculations. However, there are hardly any public information about the local drainage as we tried to acknowledge in section 2 Material and Methods (ll. 158-161). As a conservative estimate and in view of the given spatial resolution, we assumed the drainage system to be entirely inactive or malfunctioning throughout all simulations. The same rationale may explain, why other studies have abandoned the idea of implementing the HCMC drainage system in comparable studies (e.g. Scussolini et al. 2017). Last but not least, one could argue that the drainage capacity would be identical for all three considered adaptation cases and any drainage-related reduction of runoff volumes should hence be cancelled out in the subsequent comparison as we discuss in section 4 Discussions and Conclusions (ll. 320-324). To emphasize the applicability of these assumptions and underline our chosen conservative approach, we have revised our methodological explanations as follows:

*"As there is hardly any public information about the quality and condition of the existing drainage system in HCMC – except that its capacity is regularly overloaded during extreme events – all rainfall is assumed to become gradient-controlled surface runoff as is a common conservative estimate (e.g. Scussolini et al., 2017)."*

6. **In Table 1, Flood depth (dmax) is the average for all the raster cells? This seems very simple. More analysis should be done for different cells, especially considering the spatial distribution.**

It is correct that the original results section only presents mean values of flood depth ($d_{max}$) and flood duration ($T_{d>10cm}$) for all the raster cells and we have to admit that this poses the risk of losing information, especially regarding the spatial distribution of these flood proxies. Accordingly, we offer incorporating visualizations of the spatial distribution of flood depth and duration, corresponding to the NFSI in Figure 3, for all adaptation strategies in the Supplemental Material. Moreover, we have prepared an additional figure showing the relative frequency of maximum inundation depths and duration, which illustrates the hydraulic effectiveness of the discussed adaptation options in greater detail.

7. **Spatial distribution of Flood Severity (NFSI) for different cells?**

We understand this question to be directly related to the previous comment, suggesting that insights from the numerical simulations might be lost if results are solely presented in terms of overall mean values (Table 2). Although we generally share this concern and complemented our results section accordingly, we believe that Figure 3 already comprises much of the available (spatial) information about flood severity. In its current version, the illustration gives a comprehensive overview of how the assessed adaptation options would reduce the Normalized Flood Severity Index (NFSI), and thus inherently also maximum flood depth and duration, across the model domain. Corresponding to our previous response, we can offer to provide spatial data about the absolute NFSI as Supplemental Material. However, if our interpretation of this comment was incorrect, we would be grateful for additional remarks to further improve our study.

8. **This study only considers three scenarios of the mitigation scenarios, I think more analysis considering different sponge city measures would be very interesting. You will want to discuss this.**

We fully agree that it would be a valuable objective to simulate different technical solutions that put the Sponge City concept into practice. However, our conceptual approach and, more specifically, the implementation of rainwater detention in the form of an attenuated hyetograph partly rules out this kind of investigations. Although a comparison of solutions is generally possible on an analytical basis – investigating what measure would theoretically cause which attenuation – their parametrized implementation in the model would still be very similar. Given that the main goal of this study was to compare the working principles of classic flood protection and decentralized rainwater detention, the

suggested parameter study (regarding different attenuated hyetographs) would go beyond juxtaposing these two general concepts of flood adaptation. Corresponding to the reviewer's remark, we added the following paragraph in section 4 Discussions and Conclusions (ll. 326-332 and 336-342):

*"Even if a comparison of different technical solutions for this type of climate adaptation is generally possible on an analytical basis – investigating what rainwater detention measure would theoretically cause which attenuation – their parametrized implementation in the model would nearly be identical as is their general working principle. As a consequence, also the experienced flooding would be very similar in its pattern of depth and duration for all realizations of the Sponge City concept. (…) Nevertheless, the presented approach is sufficiently descriptive for a conceptual juxtaposition of the effects and performance of the two mitigation strategies under consideration and demonstrates the general working principle despite the underlying simplifications: The large-scale pumping stations comprised in the classic protection scheme reduce flood volumes along the inner-city canals and thus represent a line or even point sink within the numerical model; the implementation of the Sponge City concept, in contrast, is characterized by spatially uniform runoff attenuation, which translates to an area sink for flood volumes across the whole model domain. The aim of this study was to compare the working principles behind these two, seemingly adverse adaptation options, which is thoroughly accomplished by the employed conceptual approach."*

---

## Author Comment (AC2)

**RESPONSE TO REVIEW #2**

We highly appreciate time and effort the reviewers put into studying and reviewing our manuscript, thank you for initiating this exchange! After carefully reading and discussing your remarks, we have thoroughly revised and improved the manuscript accordingly. Please find our responses (blue) and revised text blocks (*blue, italic*) below your comments (**black, bold**). Please note that, in the meantime, we had the invaluable opportunity to visit the study site in person and subsequently incorporated some newly gathered ground truth data in the numerical modelling scheme. Moreover, we included a few minor changes that might not refer directly to specific reviewer comments, but are meant to enhance the readability and hence understanding of our approach and findings according to a native speaker.

**General Comments**

This study presents an interesting topic on low-regret climate change adaption for flood hazard in Ho Chi Minh City of Vietnam. The study has compared the effectiveness of three techniques in flood adaption strategies including: (1) a large-scale protection project using a ring dike; (2) a small-scale rainwater detention scheme; and (3) a combined application of both strategies.

We are delighted to hear that the reviewer considers the presented study an interesting topic and carefully revised our manuscript according to the provided comments. The specific points below helped us to clarify important aspects of our approach and, in the end, improved the readability and quality of the manuscript.

**The authors should consider the following specific points:**

1. Please include a flowchart of data processing in the manuscript for better understanding of the framework of this study.

Corresponding to the fourth comment of this review and in response of the first review, we first improved the readability of Figure 2 by simplifying some of the depicted elements and ensured that no information is lost during PDF conversion. In a second step, we intensively discussed the option of providing a separate flowchart to specify the individual processing steps for setting up the numerical model, but decided against it for the following reason: Beside juxtaposing the hydraulic effects of two (seemingly adverse) adaptation concepts and quantifying how flood intensities across the model domain could be reduced, the third objective of this study was to evaluate the low-regret character of these options and discuss implications in decision-making processes with all relevant stakeholders (II. 130-135). Especially this third objective requires a sound focus and well-defined research scope, which confirmed us to strictly divide the description of our methodology on the one hand,

and contextualization of results on the other, into two separate manuscripts in the same Special Issue of NHESS. The suggested flowchart of data processing forms part of the first of these companion papers and was submitted by Hoballah Jalloul et al. (in review). However, we fully understand that there is a need for more clarity in this point and added an explicit in-text reference here. This should allow (numerically) interested readers to follow up on the methodology, while maintaining a flood risk management perspective in this study. The description of the model setup (referring to Fig. 2 b) now ends as follows:

"For a more detailed explanation regarding the processing of input data as well as the calibration/validation of the employed model, please be referred to the independent publication by Hoballah Jalloul et al. (in review), which discusses the general validity of open-access data in numerical analyses more profoundly."

**2. How the Normalized Flood Severity Index (NFSI) was generated from the input parameters?**

We acknowledge the lack of clarity with regard to the Normalized Flood Severity Index (NFSI) as a consequence of our attempt to balance the readability for non-experts in the field by simplifying technical terms. We also understand that readers should get all information that are relevant to understand the implications arising from the definition of this variable and its significance for risk assessment. Accordingly, we added the following mathematical definition of the NFSI to this manuscript:

"For a given pair of coordinates, the NFSI is calculated as the product of these two conventional flood intensity proxies divided by the product of the 95 % percentiles of the same proxies as follows:

$$NFSI(x,y)(\%) = \frac{z_{max}(x,y) * DoT(x,y)}{z_{95\%}(x,y) * DoT_{95\%}(x,y)} * 100$$
(1)

where  $z_{max}(x, y)$  refers to the maximum simulated flood depth at coordinates x and y and DoT(x, y) refers to the inundation duration over the pre-defined threshold at the same coordinates."

For further details, an in-text reference should now direct the interested reader to the companion paper by Hoballah Jalloul et al. (in review), which further elaborates on the methodological details:

"A detailed explanation of the rationale behind the NFSI and its validation can be found in an independent publication by Hoballah Jalloul et al., (in review)."

**3. Line 162: Using reported data in 2010 for validation of the model is out of date. Please use the most recent field data for evaluation.**

The reviewer raises an important point here, rightly suggesting that validation data from 2010 might be outdated meanwhile. In practice, the decision for this specific data set had to be based on the co-occurrence of all boundary conditions (discharge, tidal water-levels, precipitation) with independent verification data in the form of credible reports of urban inundations at a considerable number of locations. In combination with the goal of using open-access data alone, suitable events were limited to a handful of options. While three of the available data sets (2010/2013) served as calibration data, the remaining event with the most data points (2010) was used for validation. We agree that more recent validation data would be highly desirable, but did not see any opportunity to acquire such data without unreasonable efforts. To document this predicament, we added the following explanation to the revised manuscript (II. 164-169):

"Although verification data from 2010 might seem outdated meanwhile, the practical choice for this event had to be based on the co-occurrence of all boundary conditions (incl. discharge, tidal water-levels and precipitation) with credible reports of urban inundations at a considerable number of locations. In combination with the goal of using open-access data alone, suitable events were limited to a handful of options. While three of these data sets (2010/2013) served as calibration data, the remaining event with the most data points (2010) was used for validation."

**4. Figure 2: all of the legends in this figure should be more clearly presented.**

This feedback is very helpful as it seems that illustrations originally saved in 600 dpi resolution were compressed during PDF conversion. We will make sure that all figures are provided in sufficient quality in the revised manuscript. Based on this comment, we also revisited mentioned Figure 2 and decided to simplify some of the depicted model in- and outputs for a more intuitive perception of the illustrated workflow.

---

## Author Response (AR1)

To whom it may concern,

We highly appreciate time and effort the reviewers have put into studying and reviewing our manuscript, thank you for initiating this exchange! After carefully reading and discussing the remarks, we have thoroughly revised and improved the manuscript accordingly. Please find our responses (blue) and revised text blocks (*blue, italic*) below the quoted reviewer comments (**black, bold**). Please note that, in the meantime, we also had the invaluable opportunity to visit the study site in person again. With the reviewers' comments and issues in mind, we gathered additional ground truth data and, to address the raised questions duly, incorporated them to validate and back-up the employed modelling scheme. Although the hydro-numerical results generally persist in their qualitative outcome, there have been some quantitative changes in the results section due to refined information on the pumping capacities on site. Moreover, we included a few minor modifications of the text that might not refer directly to specific reviewer comments, but are meant to enhance the readability and with it understanding of our approach and findings according to a native speaker.

With kind regards,

Leon Scheiber, Gabriel David, Mazen Hoballah Jalloul, Jan Visscher, Hong Quan Nguyen, Roxana Leitold, Javier Revilla Diez and Torsten Schlurmann
* * *
**RESPONSE TO REVIEW #1**

**General Comments**

**This paper compares the effectiveness of two adaptation strategies: (1) a large-scale flood protection scheme as currently constructed in HCMC and (2) the widespread installation of small-scale rainwater detention as envisioned in the framework of the Chinese Sponge City Program (SPM). As authors claimed, it is important to explore and evaluate potential options of hazard mitigation as flood risk is becoming a major challenge for many cities in low elevation coastal zones. The topic of this study is valuable. But the quality and innovation of the current manuscript are not satisfactory. In any case, I have a few recommendations that I believe will help the authors to clarify their contribution and improve the readability of the manuscript.**

We are very happy that the reviewer acknowledges our motivation and the intended value of the presented topic. This statement confirms our original idea to present and disseminate our simple, but generic approach to explore and evaluate hazard mitigation option to alleviate urban flood risks in low elevation coastal zones and our specific findings from HCMC to a wider audience. We are in line with the reviewer's perspective that flood exposure studies and tailor-made adaptation measures become gradually more important, especially to flood-prone communities in emerging economies and developing countries. To improve the quality and innovation of the current manuscript, we have addressed and diligently integrated the reviewer's queries and recommendations. The provided comments, indeed, helped us a lot to clarify the objectives and current limitations of our contribution and improved the readability and quality of the manuscript. Details about the specific revisions can be found below.

**Specific points are:**

1. **Most of the figures in the manuscript are very poor in quality and hard to meet the standard for this journal, such as Figs. 2, not clear enough. Generally, some of the figures are too small.**

   This feedback is very helpful as it seems that illustrations originally saved in 600 dpi resolution were compressed and therefore blurred during PDF conversion. We will make sure that all figures will be provided in original quality in the revised manuscript. Based on this comment, we also revisited mentioned Fig. 2 and decided to simplify some of the depicted model in- and outputs for a more intuitive perception of the illustrated workflow.

2. **How to simulate the small-scale rainwater detention in the HEC-RAS model? What is the limitation or uncertainty?**

   We are happy that the reviewer pointed out this lack of clarity. Following the previous comment, we tried to improve the readability of figures including our illustration of the attenuated hyetograph (Fig. 2 c right), which allows us to simulate small-scale rainwater detention by means of a parametrization. We hope that, accordingly, the explanations given in section 2 Material and Methods (ll. 218-227) should now give a sufficient explanation of the rationale behind this way of implementation. Instead, we revised section 4 Discussions and Conclusions (ll. 380-391) adding further discussion of the limitations/uncertainties inherent to this approach. The paragraph now reads as follows:

*"Additional limitations arise from the parametrization of rainwater detention in the form of an attenuated hyetograph (cf. section 2 Material and Methods). (…) The estimation of the roofed area from satellite imagery and corresponding detention capacities entails additional uncertainties with respect to the assumed runoff attenuation. In reality, the actual shape of this hyetograph depends on a multitude of factors including technical details about the individual solutions (how much storage volume per unit) as well as the degree of implementation (how many units per area). Nevertheless, the presented approach is sufficiently descriptive for a conceptual juxtaposition of the effects and performance of the two mitigation strategies under consideration and demonstrates the general working principles despite the underlying simplifications: (…)"*

3. **I am not convinced by the model setup given limited information, more information for the validity of flood simulation by HEC-RAS should be described in the paper.**

This comment addresses a concern, which has already been much-discussed within our group before submission. In the current version of the manuscript, explanations regarding the numerical setup were deliberately kept short to be readily accessible for non-experts in the field of hydraulics and water resources. The same group of authors currently aims to publish, in parallel, the more detailed aspects of the flood simulation in a companion paper submitted by Scheiber et al. (in review) in the same Special Issue of NHESS. This second manuscript not only addresses all specificities about the acquisition and processing of input data and the calibration/validation of the model, but also elaborates on the general validity and performance of open-access data in any numerical flood risk analyses. The methodology is meant to enhance understanding and build capacities to create sound adaptation strategies. In order to keep the focus and objectives of our work as distinct as possible, we decided to separate the two studies and consequently reduced methodological details in the current manuscript to a minimum. Nevertheless, we added an explicit in-text reference here to improve the balance between both publications. This allows readers with a modelling background to follow up on the methodology, while maintaining a risk management perspective within this study. The description of the model setup (referring to Fig. 2 b) has been revised and now reads as follows:

*"For a more detailed explanation regarding the processing of input data as well as the calibration/validation of the employed model, please be referred to the independent publication by Scheiber et al. (in review), which discusses the general validity of open-access data in numerical analyses more profoundly."*

4. **What is "Flood Severity Index (NFSI) ", how is it measured?**

Similar to the previous comment, we acknowledge the lack of clarity with regard to the Normalized Flood Severity Index ($I_{NFS}$) as a consequence of our attempt to balance the readability for non-experts in the field by simplifying technical terms. We also understand that readers should get all information that are relevant to understand the implications arising from the definition of this variable and its significance for risk assessment. Accordingly, we added the following mathematical definition of the $I_{NFS}$ to this manuscript:

*"For a given pair of coordinates, the INFS is calculated as the product of these two conventional flood intensity proxies divided by the product of the 95th percentiles of the same proxies as follows:*

$$I_{NFS}(x,y)(\%) = \frac{d_{max}(x,y) * T_{d>10cm}(x,y)}{d_{max,95\%}(x,y) * T_{d>10cm,95\%}(x,y)} * 100 \qquad (1)$$

*where $d_{max}$ (x,y) refers to the maximum simulated flood depth over time at coordinates x and y and $T_{d>10cm}$ (x,y) refers to the inundation duration over the pre-defined threshold of 10 cm at the same coordinates. Using the 95th spatial percentiles in the denominator eliminates the distorting effects of potential numerical artefacts. As a qualitative first estimate, this easy-to-apply index emphasizes those areas, which are exposed to significant flooding over a significant time and, thus, are expected to experience the most severe damage across a given study area."*

For further details, an in-text reference should now direct the interested reader to the companion paper by *Scheiber et al. (in review)*, which further elaborates on the methodological details:

*"A detailed explanation of the rationale behind the $I_{NFS}$ and its validation can be found in an independent publication by Scheiber et al., (in review)."*

5. **Drainage capacity should be considered for the flood simulation.**

The reviewer addresses an important limitation, here. We generally agree that a direct implementation of this detail would be highly desirable, as it could add further credibility to the numerical setup and robustness to subsequent calculations. However, there are hardly any public information about the local drainage as we tried to acknowledge, but now further emphasize in section 2 Material and Methods (ll. 178-170):

*"As there is hardly any public information about the quality and condition of the existing drainage system in HCMC – except that its capacity is regularly overloaded during extreme events – all rainfall is assumed to become gradient-controlled surface runoff as is a common conservative estimate (e.g. Scussolini et al., 2017)."*

As a conservative estimate and in view of the given spatial resolution, we assumed the drainage system to be entirely inactive or malfunctioning throughout all simulations. The same rationale may explain, why other studies have abandoned the idea of implementing the HCMC drainage system in comparable studies (e.g. Scussolini et al. 2017). Last but not least, one could argue that the drainage capacity would be identical for all three considered adaptation cases and any drainage-related reduction of runoff volumes should hence be cancelled out in the subsequent comparison of simulations as we discuss in section 4 Discussions and Conclusions (ll. 344-347).

6. **In Table 1, Flood depth (dmax) is the average for all the raster cells? This seems very simple. More analysis should be done for different cells, especially considering the spatial distribution.**

It is correct that the original results section only presents mean values of flood depth ($d_{max}$) and flood duration ($T_{d>10cm}$) for all the raster cells and we have to admit that this poses the risk of losing information, especially regarding the spatial distribution of these flood proxies. Accordingly, we incorporated visualizations of the spatial distribution of reductions in flood depth and duration, corresponding to the $I_{NFS}$ in Figure 3, for all adaptation strategies in the Supplemental Material. Moreover, we have prepared an additional figure showing the relative frequency of maximum inundation depths, duration and $I_{NFS}$, which illustrates the hydraulic effectiveness of the discussed adaptation options in greater detail.

7. **Spatial distribution of Flood Severity (NFSI) for different cells?**

We understand this question to be directly related to the previous comment, suggesting that insights from the numerical simulations might be lost if results are solely presented in terms of overall mean values (Table 2). Although we generally share this concern and complemented our results section accordingly, we believe that Figure 3 already comprises much of the available (spatial) information about flood severity. In its current version, the illustration gives a comprehensive overview of how the assessed adaptation options would reduce the Normalized Flood Severity Index ($I_{NFS}$), and thus inherently also maximum flood depth and duration, across the model domain. Corresponding to our previous response, we provide spatial data about the absolute $I_{NFS}$ as Supplemental Material. However, if our interpretation of this comment was incorrect, we would be grateful for additional remarks.

8. **This study only considers three scenarios of the mitigation scenarios, I think more analysis considering different sponge city measures would be very interesting. You will want to discuss this.**

We fully agree that it would be a valuable objective to simulate different technical solutions that put the Sponge City concept into practice. However, our conceptual approach and, more specifically, the implementation of rainwater detention in the form of an attenuated hyetograph partly rules out this kind of investigations. Although a comparison of solutions is generally possible on an analytical basis – investigating what measure would theoretically cause which attenuation – their parametrized implementation in the model would still be very similar. Given that the main goal of this study was to compare the working principles of classic flood protection and decentralized rainwater detention, the suggested parameter study (regarding different attenuated hyetographs) would go beyond juxtaposing these two general concepts of flood adaptation. Corresponding to the reviewer's remark, we added the following paragraph in section 4 Discussions and Conclusions (ll. 351-355 and 359-366):

*"Even if a comparison of different technical solutions for this type of climate adaptation is generally possible on an analytical basis – investigating what rainwater detention measure would theoretically cause which attenuation – their parametrized implementation in the model would nearly be identical, as is their general working principle. As a consequence, also the experienced flooding would be very similar in its pattern of depth and duration for all realizations of the Sponge City concept.(…) Nevertheless, the presented approach is sufficiently descriptive for a conceptual juxtaposition of the effects and performance of the two mitigation strategies under consideration and demonstrates the general working principles despite the underlying simplifications: The large-scale pumping stations comprised in the classic protection scheme reduce flood volumes along the inner-city canals and thus represent a line or even point sink within the numerical model; the implementation of the Sponge City concept, in contrast, is characterized by spatially uniform runoff attenuation, which translates to an area sink for flood volumes across the whole model domain. The aim of this study, i.e. to compare the working principles behind these two, seemingly adverse adaptation options, is thoroughly accomplished by the employed conceptual approach."*

**General Comments**

**This study presents an interesting topic on low-regret climate change adaption for flood hazard in Ho Chi Minh City of Vietnam. The study has compared the effectiveness of three techniques in flood adaption strategies including: (1) a large-scale protection project using a ring dike; (2) a small-scale rainwater detention scheme; and (3) a combined application of both strategies.**

We are delighted to hear that the reviewer considers the presented study an interesting topic and carefully revised our manuscript according to the provided comments. The specific points below helped us to clarify important aspects of our approach and, in the end, improved the readability and quality of the manuscript.

**The authors should consider the following specific points:**

**1. Please include a flowchart of data processing in the manuscript for better understanding of the framework of this study.**

Corresponding to the fourth comment of this review and in response of the first review, we first improved the readability of Figure 2 by simplifying some of the depicted elements and ensured that no information is lost during PDF conversion. In a second step, we intensively discussed the option of providing a separate flowchart to specify the individual processing steps for setting up the numerical model, but decided against it for the following reason: Beside juxtaposing the hydraulic effects of two (seemingly adverse) adaptation concepts and quantifying how flood intensities across the model domain could be reduced, the third objective of this study was to evaluate the low-regret character of these options and discuss implications in decision-making processes with all relevant stakeholders (ll. 138-143). Especially this third objective requires a sound focus and well-defined research scope, which confirmed us to strictly divide the description of our methodology on the one hand, and contextualization of results on the other, into two separate manuscripts in the same Special Issue of NHESS. The suggested flowchart of data processing forms part of the first of these companion papers and was submitted simultaneously by Scheiber et al. (in review). However, we fully understand that there is a need for more clarity in this point and added an explicit in-text reference here. This should allow (numerically) interested readers to follow up on the methodology, while maintaining a flood risk management perspective in this study. The description of the model setup (referring to Fig. 2 b) now ends as follows:

*"For a more detailed explanation regarding the processing of input data as well as the calibration/validation of the employed model, please be referred to the independent publication by Scheiber et al. (in review), which discusses the general validity of open-access data in numerical analyses more profoundly."*

**2. How the Normalized Flood Severity Index (NFSI) was generated from the input parameters?**

We acknowledge the lack of clarity with regard to the Normalized Flood Severity Index ($I_{NFS}$) as a consequence of our attempt to balance the readability for non-experts in the field by simplifying technical terms. We also understand that readers should get all information that are relevant to understand the implications arising from the definition of this variable and its significance for risk assessment. Accordingly, we added the following mathematical definition of the $I_{NFS}$ to this manuscript:

*"For a given pair of coordinates, the INFS is calculated as the product of these two conventional flood intensity proxies divided by the product of the 95th percentiles of the same proxies as follows:*

$$I_{NFS}(x,y)(\%) = \frac{d_{max}(x,y) * T_{d>10cm}(x,y)}{d_{max,95\%}(x,y) * T_{d>10cm,95\%}(x,y)} * 100 \quad (1)$$

*where $d_{max}$ (x,y) refers to the maximum simulated flood depth over time at coordinates x and y and $T_{d>10cm}$ (x,y) refers to the inundation duration over the pre-defined threshold of 10 cm at the same coordinates. Using the 95th spatial percentiles in the denominator eliminates the distorting effects of potential numerical artefacts. As a qualitative first estimate, this easy-to-apply index emphasizes those areas, which are exposed to significant flooding over a significant time and, thus, are expected to experience the most severe damage across a given study area."*

For further details, an in-text reference should now direct the interested reader to the companion paper by Scheiber et al. (in review), which further elaborates on the methodological details:

*"A detailed explanation of the rationale behind the $I_{NFS}$ and its validation can be found in an independent publication by Scheiber et al., (in review)."*

3. **Line 162: Using reported data in 2010 for validation of the model is out of date. Please use the most recent field data for evaluation.**

The reviewer raises an important point here, rightly suggesting that validation data from 2010 might be outdated meanwhile. In practice, the decision for this specific data set had to be based on the co-occurrence of all boundary conditions (discharge, tidal water-levels, precipitation) with independent verification data in the form of credible reports of urban inundations at a considerable number of locations. In combination with the goal of using open-access data alone, suitable events were limited to a handful of options. While three of the available data sets (2010/2013) served as calibration data, the remaining event with the most data points (2010) was used for validation. We agree that more recent validation data would be highly desirable, but did not see any opportunity to acquire such data without unreasonable efforts. To document this predicament, we added the following explanation to the revised manuscript (ll. 173-178):

*"Although verification data from 2010 might seem outdated meanwhile, the practical choice for this event had to be based on the co-occurrence of all boundary conditions (incl. discharge, tidal water-levels and precipitation) with credible reports of urban inundations at a considerable number of locations. In combination with the goal of using open-access data alone, suitable events were limited to a handful of options. While three of these data sets (2010/2013) served as calibration data, the remaining event with the most data points (2010) was used for validation."*

4. **Figure 2: all of the legends in this figure should be more clearly presented.**

This feedback is very helpful as it seems that illustrations originally saved in 600 dpi resolution were compressed during PDF conversion. We will make sure that all figures are provided in sufficient quality in the revised manuscript. Based on this comment, we also revisited mentioned Figure 2 and decided to simplify some of the depicted model in- and outputs for a more intuitive perception of the illustrated workflow.

REFERENCES:

Scheiber, L., Hoballah Jalloul, M., Jordan, Christian, Visscher, Jan, and Schlurmann, T.: The Potential of Open-Access Data for Flood Estimations: Uncovering Inundation Hotspots in Ho Chi Minh City, Vietnam, through a Normalized Flood Severity Index, Nat. Hazards Earth Syst. Sci., in review, https://doi.org/10.5194/nhess-2022-238.

Scussolini, P., van Tran, T. T., Koks, E., Diaz-Loaiza, A., Ho, P. L., and Lasage, R.: Adaptation to Sea Level Rise: A Multidisciplinary Analysis for Ho Chi Minh City, Vietnam, Water Resour. Res., 53, 10841–10857, https://doi.org/10.1002/2017WR021344, 2017.

---

## Author Response (AR2)

**Dear authors,**

**Thanks for providing the well updated manuscript, which has been reviewed again by three reviewers. All reviewers' comments are minor issues that I believe the authors can well address them. In particular, I remind the authors to discuss the different precipitation scenarios and the heterogeneities when applying small-scale rainwater detention approach. Based on these, I suggest an overall minor revision of the paper before it can be accepted for publishing.**

Dear Special Issue Editors, dear Dr. Yang,

We are delighted that our efforts to clarify the original idea and motivation for our study in order to improve the initial manuscript are acknowledged. Based on the newly provided comments, we have undertaken another revision, specifically elaborating on our choice of a 'most representative' precipitation scenario and the impact of intra-city heterogeneities. Thanks to your and the reviewers' comments, the quality of the manuscript could further be improved.

With kind regards,
Leon Scheiber, Gabriel David, Mazen Hoballah Jalloul, Jan Visscher, Hong Quan Nguyen, Roxana Leitold, Javier Revilla Diez and Torsten Schlurmann
* * *
**RESPONSE TO REPORT #1**

**All of my comments were addressed properly. I recommend this paper be accepted for publication. The important reference this paper has cited is a paper under review process which is Scheiber et al., (in review). Please check again how to cite an unpublished paper.**

We are happy to hear that our revisions meet the reviewer's expectations. Moreover, we are thankful for this remaining technical caveat, which concerned us as well. To gain clarity, we have asked the editorial office of NHESS for advice: we were informed that, in our specific case, both companion papers can be accepted simultaneously, so that their 'under review' status will be changed by the production team. According to this statement, there should not be further technical constraints.

**RESPONSE TO REPORT #2**

**No more comments (Accept as is).**

We are more than happy that we were able to compose a manuscript which, according to the reviewer's perception, can be accepted as is. We thank the reviewer for his/her efforts that were instrumental in improving the initial submission.

**RESPONSE TO REPORT #3**

**This study evaluated and discussed the effectiveness of two flood adaption strategies in Ho Chi Minh City, including the large-scale ring dike protection scheme and decentralized small-scale rainwater detention. Significant reductions of flood hazard, as indicated by Normalized Flood Severity Index (INFS) which considering flood depth and duration, have been revealed for both strategies and their combination. The manuscript is well written, with comprehensive and meaningful analysis and discussions on the implementation of different climate change adaption strategies for HCMC, which also enlightening other areas such as LECZ and ASEAN. I enjoyed reading it. Though I believe the manuscript is publishable in the present form, there are few minor comments for the authors consideration.**

**1) It seems that certain precipitation scenario was used for the HCMC flood case. The uncertainties of modelling results shall be considered when comparing the effectiveness of different strategies. There are at least two issues related to the choice of precipitation scenario. First, the effectiveness of two strategies at different scales may be aimed for different kinds of flood events, for example, storm surges or pluvial floods. However, certain precipitation scenario shall not be able to reflect such differences. Moreover, how the effectiveness would change with the scale of flood is not clear enough neither.**

The reviewer makes a valid point here, stating that the decision for a specific precipitation scenario impacts the calculated effectiveness and overall reliability of different adaptation strategies. We agree that our assumptions and the resulting uncertainties should be documented. We revised the manuscript accordingly. The corresponding discussion section (chapter 4) now reads as follows (ll. 329-338):

*"Additional limitations arise from the definition of a design storm and from the parametrization of rainwater detention in the form of an attenuated hyetograph (cf. section 2 Material and Methods). The precipitation boundary was determined in conjunction with a systematic*

*mapping of local policies and adaptation guidelines. It follows a 180 min design storm specified by an official decree enforced in decision 752/QD TTg of the HCMC government. This duration may be seen as a balance point: while the large-scale ring dike and pumping stations are constructed to cope with extraordinary precipitation volumes (mostly in combination with spring tides), the suggested detention measures aim at complementing the local drainage system during shorter cloudburst events. The same applies to the return period, which certainly has a considerable impact on the effectiveness and technical limitation of the compared strategies. Nevertheless, the commitment to this specific design storm has to be seen in a row of conceptual assumptions that were necessary to undertake a direct comparison of these hydraulically unlike adaptation options."*

**2) The intra-city heterogeneities shall also be important, especially for the small-scale rainwater detention approach. One advantage of the decentralized rainwater detention is the feasibility to be applied to local scale, for example, the flood prone areas due to the unfavoured topography, land use, drainage system etc. However, it is a bit hard to identify such information from the "more uniformly distributed" flood reduction of the small-scale rainwater detention strategy.**

We agree with the reviewer that it is a major advantage of small-scale detention measures to be implemented in a decentralized manner that is based on people-centered, i.e. bottom-up decisions. This makes the approach applicable to many regions, including those where a large-scale protection scheme is ineffective. But then, the detention measures can also relieve pressure from the downstream parts of the drainage system independent of whether they are installed in flood-prone areas or not. For this conceptual study, it was therefore decided to consider this type of adaptation to be "uniformly distributed" rather than making assumptions about the local feasibility and most effective placement in specific boroughs or neighborhoods. After all, such in-depth considerations about intra-city heterogeneities, unfortunately, can neither be proven nor evaluated at the available level of detail (e.g. about the drainage system). However, in order to sensitize readers to the existence of presumable heterogeneities and to clarify the rationale behind our uniform parametrization, we have added the following lines to the discussion of assumptions regarding our parametrization scheme (ll. 343-346):

*"Even if presumable heterogeneities in the realization of rainwater detention measures are neglected by this uniform parametrization, it still allows for the fact that subsidized micro-scale solutions may be implemented in a people-centered, i.e. bottom-up approach. In the first place, this new paradigm in adaptation can complicate official monitoring, but may render supervision by local authorities unnecessary in the end."*